# How People Manage Relationship Problems: An Exploratory Study in the Greek Cultural Context

**DOI:** 10.3390/bs15020218

**Published:** 2025-02-15

**Authors:** Menelaos Apostolou, Anthi Samara, Timo Juhani Lajunen

**Affiliations:** 1Department of Social Sciences, University of Nicosia, 46 Makedonitissas Avenue, Nicosia 1700, Cyprus; apostolou.m@unic.ac.cy (M.A.); samara.a7@live.unic.ac.cy (A.S.); 2Department of Psychology, Norwegian University of Science and Technology, 7491 Trondheim, Norway

**Keywords:** relationship problems, managing relationship problems, mating strategies, mating

## Abstract

Problems frequently arise in intimate relationships, and people need to find ways to manage them if they want their relationships to continue. The purpose of the current research was to identify the strategies that people use to manage relationship problems, employing a mixed-methods approach, specifically, in a sample of 224 Greek-speaking participants. Study 1 employed qualitative research methods and identified 93 actions that people perform to manage relationship problems. Study 2 employed quantitative research methods in a sample of 526 Greek-speaking participants and classified these actions into three main strategies. The most likely to be used main strategy was “Solve the problem”, where participants attempted to solve the problem by discussing and negotiating it with their partners, making compromises, improving physical intimacy, and avoiding acting in the heat of the moment. The second most likely to be used main strategy was “External input”, where participants aimed to obtain advice and support from others, including friends, relatives, and psychologists. The least likely to be used strategy was “Avoid the problem”, where participants attempted to manage their problems by ignoring or avoiding them. Participants indicated that they were likely to use a combination of strategies to manage their problems. Men and women were generally in agreement over which strategy to use, while age effects were found for several strategies.

## 1. Introduction

Intimate relationships are frequently plagued with problems that people need to deal with ([3]; [13]; [21]). One way to address these issues is to terminate the relationship, perhaps seeking one with fewer problems. Yet, as problem-free relationships are rare or nonexistent, people need to manage these problems if they wish to stay mated in the long run. The purpose of the current study is to identify the different strategies that individuals use for this purpose. We will begin by discussing the root causes of relationship problems.

### 1.1. The Nature of Relationship Problems

When engaging in mating, individuals employ different strategies, i.e., acts and behaviors that enable them to increase reproductive success or fitness ([8]; [19]). People engage in intimate relationships with others when they assess that they follow compatible mating strategies and thus have common fitness interests. However, because the two parties who form an intimate relationship are not genetically identical, the strategy that one party employs to increase their fitness does not necessarily serve the fitness interests of the other party, leading to conflict between the two ([6]; [9], [10]).

Extra-pair mating constitutes a good example of such conflict; individuals can increase their reproductive success by having families with long-term mates, but they can increase their reproductive success further by having extra-pair relationships. For instance, a man can have additional offspring with extra-pair mates, while a woman can secure better genes than those of her legitimate partner ([7]). Yet, an extra-pair mating strategy would decrease the legitimate partner’s reproductive success. For example, men risk raising other men’s children without being aware of it, and women risk losing their partner’s investment to other women. Diverging interests are a main source of relationship problems. In the example above, people would attempt to have extra-pair relationships while at the same time preventing their partners from doing so, leading to conflict between the two. For instance, one partner may insist on always accompanying the other when going out ([2]; [11]), something that the other may not always find acceptable as it reduces their freedom.

Even if the interests of the parties involved in the relationship closely align, strain may still arise due to incompatible traits. Intimate partners may, for instance, differ in terms of personality traits ([29]). For example, one partner may be extraverted and the other introverted, which could result in relationship strain as partners would disagree on how to spend their free time. The evolutionary mismatch problem may be another source of relationship problems. In particular, all human adaptations have predominantly been shaped in the ancestral environment and may not be effective in dealing with the demands of the modern one if this is very different from the ancestral environment ([16]; [28]). Evolutionary scholars have not provided a systematic study on how long-term intimate relationships in ancestral human societies differed from contemporary ones, yet there are reasons to believe that there were considerable differences. For instance, there are reasons to believe that much of human evolution occurred in a context where human rights were not well protected, and men held dominance over women ([12]; [27]). Consequently, it is possible that men, who are on average larger and stronger than women, evolved a tendency to use physical aggression to impose their will on their wives. In a contemporary Western context, where human rights are well protected and women are not coerced into relationships, such male tendencies would be problematic, potentially leading to relationship strain. In this respect, the root of relationship strain is the mismatch problem rather than conflict between the sexes. We can now proceed to examine the problems that people in intimate relationships face.

### 1.2. Problems in Intimate Relationships

Studies on divorce give us an idea about what problems people face in their intimate relationships. To begin with, one study employed a sample of 153 newly separated but not yet divorced individuals in the United States and found that communication difficulties, value conflicts, and boredom were common sources of marital dissatisfaction, while physical abuse, drinking, and drug abuse were less common sources ([4]). Another study examined the interviews of 208 divorced individuals in the United States and found that infidelity was the most commonly reported cause, followed by incompatibility, drinking or drug use, and growing apart ([1]). In the same vein, a different study in the USA employed a sample of 886 divorcing individuals and found that the most common reasons for divorce were growing apart, communication problems, not enough attention from the spouse, and infidelity ([21]). A more recent study employed a sample of 1403 participants from China and Greece who were in an intimate relationship and asked them what difficulties they faced in keeping an intimate relationship ([3]). Among the most common ones were clinginess, long work hours, and lack of personal time and space. Other difficulties included character issues, lack of effort, violence and addictions, fading enthusiasm, and infidelity. Overall, people face a diverse set of problems in their relationships that call for a diverse array of strategies to manage them.

### 1.3. How People Manage Relationship Problems

Forming long-term intimate relationships is not a recent phenomenon, as there are reasons to believe that across different cultures and times, people have formed such relationships. In particular, anthropological and historical evidence indicates that marriage was practiced in all contemporary and ancestral societies of which we have some knowledge ([18]). There are also reasons, including the conflict between the sexes, to believe that these relationships have not been problem-free. It follows that, during human evolutionary time, to be effective in forming lasting relationships, people had to manage relationship problems somehow. In this respect, we expect that people have evolved a capacity to deploy a battery of strategies that would enable them to succeed in this endeavor.

Some of the arguments that we will make next are supported by research on conflict resolution. However, it is important to note that conflict resolution is different from managing relationship problems. Many, if not most, couples do not completely solve their problems; instead, they find ways to live with them. One reason for this is that not all problems can be fully resolved, and many problems can only be solved to a certain degree. Therefore, while research on conflict resolution is informative, it is not sufficient for understanding the different strategies people use to manage the problems they face in a relationship.

The most obvious strategy for managing relationship problems is to attempt to address them when they arise. One way is to discuss them with one’s partner, attempting to find a mutually acceptable solution. Research on conflict resolution between couples has found discussing and negotiating the problem with one’s partner to be one of the most common strategies ([5]; [20]; [26]; [31]; [33]). Effective negotiations require some degree of compromise, which is a common strategy in conflict resolution ([26]; [33]).

The second strategy we propose that individuals use to manage relationship issues is to seek the input of others. In particular, as people are social species that have evolved to live in groups, they discuss their problems with others, seeking their input and advice ([17]). This could potentially be an effective strategy, as others may have more experience with relationship problems, or because they are not involved in the relationship, they may see the issues more objectively. Moreover, people can observe others who manage relationship difficulties effectively and adopt some of the strategies they use. Additionally, others may provide emotional support that would enable them to manage relationship problems better. Furthermore, in contemporary societies, there are professionals such as counseling psychologists who specialize in helping people manage relationship problems. Accordingly, people may also consult such experts for advice and support.

A third strategy for managing relationship problems would be to ignore or avoid them. Research on conflict resolution finds that avoidance is one strategy that people use to resolve conflicts ([13]; [31]). As most problems usually do not go away when one ignores them, such a strategy may not seem very effective at first. Yet, it may be effective when the problem causes little disturbance to the relationship and/or cannot be easily resolved. For instance, people’s partners may have gained some extra weight, which makes them less attractive to them. Nevertheless, their partners may be very sensitive to this issue, or this increase in weight may have arisen from a health problem that makes it difficult for their partners to control their weight. Thus, people may be better off ignoring the problem than attempting to resolve it (by asking, for instance, their partner to lose weight).

A fourth strategy for managing relationship problems would be to seek the things that the legitimate partner does not provide in extra-pair partners. For instance, a common difficulty in long-term relationships is poor sex ([3]), so people may look for extra-pair sexual outlets. Parallel relationships are common, but given their secretive nature, it is difficult to say how common they are ([32]). Despite its potential benefits, this strategy has obvious costs, including the risk of their partner detecting their infidelity or retaliating, which suggests that it would not be the preferred one for managing relationship problems.

### 1.4. The Current Study

Previous research has addressed the question of how people attempt to solve relationship conflicts ([5]). On the other hand, the present research attempts to address the question of how people manage relationship problems. Our question is also broader in the sense that we refer to problems and not conflicts. Having conflicts with one’s partner is a problem in a relationship, but not all relationship problems involve conflict (see above).

We have argued that people would use at least four strategies for managing relationship problems: attempting to solve the problem, seeking the input of others, avoiding the problem, and forming extra-pair relationships. To the best of our knowledge, this is the first study in this area, and given the complexity of the phenomenon in question, our research was designed to be exploratory. That is, we cannot predict a priori all the strategies that people use to manage relationship problems. Accordingly, we adopted a mixed-methods design. In particular, in Study 1, we employed qualitative research methods to identify an inclusive list of acts that people use to manage relationship problems. In Study 2, we employed quantitative research methods to classify these acts into broader strategies. Moreover, in Study 2, we aimed to examine whether there were sex, age, and relationship status differences in the likelihood to adopt these strategies without making any directional hypotheses.

## 2. Study 1

### 2.1. Materials and Methods

#### 2.1.1. Participants

The study was conducted at a private university in the Republic of Cyprus and received ethics approval from the institution’s ethics board. Participants were recruited by sharing the study link on social media platforms (Facebook and Instagram) with individuals residing in Greece and the Republic of Cyprus, and by forwarding it to students and colleagues, who were encouraged to share it further. The only requirement for participation was being at least 18 years old. Participation was voluntary, with no monetary or other incentives provided. Data collection took place in January 2024. To enhance validity, we excluded participants who were currently single and had never been in a relationship before. Thus, the final sample consisted of 224 Greek-speaking individuals (117 women, 107 men). The mean age for women was 32.4 years (*SD* = 11.1), and for men it was 34.3 years (*SD* = 11.0). Regarding relationship status, 18.3% of participants were single, 78.6% were in an intimate relationship, and 3.1% indicated their relationship status as “other”.

#### 2.1.2. Materials

The study was conducted in Greek, created using Google Forms, and administered online. It consisted of two parts. In the first part, participants were asked to “Write down some ways that you believe have helped you manage problems in your current or previous relationships”, with space provided for their responses. The second part collected demographic information, including sex, age, and relationship status. Participants were also asked if they had ever been in an intimate relationship in the past (yes, no).

### 2.2. Analysis and Results

For data analysis, we enlisted two independent graduate students (one male and one female). Each assistant reviewed the responses and compiled a list of acts associated with managing relationship problems. They were instructed to exclude responses with unclear or vague wording. After processing approximately 30% of the responses, the assistants compared their lists and then proceeded to process the remaining responses. Two lists of acts, one from each assistant, were produced and compared. Agreement was found for most items. In cases of disagreement, the authors were consulted, and a final list of acts was agreed upon by all parties involved. A total of 93 acts were identified and are presented in Table 1.

## 3. Study 2

### 3.1. Materials and Methods

#### 3.1.1. Participants

We aimed to obtain a diverse sample of participants, so we adopted different recruitment strategies. Specifically, we forwarded the study link to students and colleagues, requesting them to share it further. In addition, we promoted the link on social media platforms, including Facebook and Instagram, targeting participants residing in Greece and the Republic of Cyprus. Moreover, we created a QR code leading to our study and recruited participants in cafés and malls in the city of Nicosia in the Republic of Cyprus. Data collection took place in March and April 2024.

To increase validity, we excluded the responses of eight participants who indicated that they were currently single and had not been in an intimate relationship in the past. In total, 526 Greek-speaking participants (272 women, 251 men, and three participants who did not indicate their sex) took part. The mean age of women was 33.7 years (*SD* = 11.1), and the mean age of men was 32.9 years (*SD* = 12.5). Moreover, 68.9% of the participants indicated that they were in an intimate relationship, 28.6% were single, and 2.5% indicated their relationship status as “other”.

#### 3.1.2. Materials

The study was conducted online, the questionnaire was in Greek, created using Google Forms, and consisted of two parts. In the first part, participants were given the following scenario: “Below you will find some ways that people use to manage relationship problems. When you are in a relationship, indicate how likely you are to do each of the following in order to manage such a problem”. Subsequently, they were given the 93 items identified in Study 1 to rate on the following scale: 1—Not at all likely to do it, 5—Very likely to do it. The order of presentation of the items was randomized across participants. In the second part, demographic information was collected, including sex (man, woman, other), age, and relationship status (single, in an intimate relationship, other). Participants were also asked if they had been in an intimate relationship before (yes, no).

#### 3.1.3. Data Analysis

To classify the 93 actions into broader factors or strategies, we applied exploratory factor analysis using principal axis factoring for factor extraction with direct oblimin rotation. To determine the number of factors to retain, we performed parallel analysis ([23]). Additionally, to decide how many items to retain in each factor, we used the 0.30 cutoff point. Moreover, we conducted a series of MANCOVA tests to identify significant associations. Specifically, for each identified strategy, the actions within that strategy were entered as dependent variables. Sex and relationship status were entered as the independent categorical variables, while age was entered as a continuous independent variable.

### 3.2. Results

#### 3.2.1. Factor Structure

Parallel analysis indicated a 10-factor solution. The KMO was 0.88, indicating that our sample was of a very good standard for exploratory factor analysis to be performed. Initial analysis showed that some items had factor loadings below the 0.30 cutoff point. To address this issue, we conducted a revised analysis by excluding the item with the lowest factor loading and repeated the procedure until a factor solution emerged where all items had loadings of at least 0.30. In total, nine items were removed, which are listed in Appendix A. The internal consistency (Cronbach’s alpha) ranged from 0.66 to 0.91 (Table 1).

The first strategy to emerge was “Communication and problem-solving with partner”, where people would discuss relationship problems with their partners, raise their concerns, and express their feelings. One facet of this strategy was to understand one’s partner’s point of view. This strategy also involved an attempt to understand the cause of the problems people faced in their relationship. The second strategy to emerge was “External input and other relationships observation”, where people discussed their relationship problems with and asked the advice of friends and relatives, especially those involved in long-term intimate relationships. One facet of this strategy was for people to learn how to manage these problems by observing relationships that worked well and relationships that have failed. In the “Concessions and compromises” strategy, people managed relationship problems by stepping back, compromising, apologizing, and taking responsibility for things they did not do well. This strategy also involved being more tolerant and forgiving of one’s partner. The next strategy to emerge was the “Reflective response”, where people avoided reacting to an argument in the heat of the moment or when angry, trying to stay calm.

To manage relationship problems, people would also “Consult an expert”, such as a psychologist, or engage in couples therapy. One facet of this strategy was to read more on this topic. Moreover, in the “Intimacy enhancement efforts” strategy, people would have more sexual and physical contact with their partners and spend more time with them. In the “Spiritual coping” strategy, people managed their relationship problems by turning to religion and discussing them with their spiritual advisor. Having a “Parallel relationship” constitutes another way of managing relationship problems. One facet of this strategy involved, for a period of time, people distancing themselves from the relationship. In the “Self-care actions” strategy, individuals would do things for themselves, including activities and physical exercise that give them pleasure. Finally, in the “Avoidance” strategy, people managed their relationship problems by not thinking about or ignoring them.

#### 3.2.2. Likelihood of Use

To identify the strategies that were more likely to be used, we calculated the means and standard deviations for each one and placed them in a hierarchical order in Table 2. Moreover, for each strategy, we calculated the percentage of participants who gave a mean score above “3”. Given our scale, this percentage would indicate the number of participants who were highly likely to employ the strategy in question.

The most likely to be used strategy was “Communication and problem-solving with partner”, with more than 96% of the participants indicating a high likelihood to use it. Next were the “Intimacy enhancement efforts” and the “Concessions and compromises” strategies, with 81.3% and 84.5%, respectively, of the participants indicating a high likelihood to use them. The “Reflective response”, the “Self-care actions”, and the “External input and other relationships observation” were in the middle of the hierarchy, while the “Avoidance” and the “Consult an expert” were below the middle. The “Parallel relationship” and the “Spiritual coping” were the least likely to be used strategies, with 5.4% and 15.5%, respectively, of the participants indicating a high likelihood to use them.

Furthermore, we calculated the percentage of individuals who were highly likely to use a combination of strategies. In particular, if participants gave a mean score above three in two strategies, they were classified as highly likely to employ two strategies. The results are summarized in Figure 1, where we can see that most participants would use between three and seven strategies.

#### 3.2.3. Significant Effects of Sex, Age, and Relationship Status

In total, 10 MANCOVA tests were performed, and to reduce the probability of committing a Type I error, we applied Bonferroni correction for alpha inflation, setting the alpha level to 0.005 (0.050/10). Accordingly, the reader should not consider any results above this level as significant. The results are summarized in Table 2.

We can see that there were no significant main effects of sex for any of the identified strategies. It is worth noting that for the “Consult an expert” strategy, the effect approached the significance level, with women giving higher scores than men. In most cases, there was a significant main effect of age. As indicated by the effect size, the largest effect was for the “Communication and problem-solving with partner” strategy, with a negative coefficient indicating that older participants were less likely to use this strategy than younger ones. The second largest effect was for the “Consult an expert” strategy, with a positive coefficient indicating that older participants were more likely to use this strategy than younger ones. Relationship status was not significant for any of the identified strategies.

Moreover, we examined whether there were sex, age, and relationship status effects on the number of strategies participants were highly likely to use (see Figure 1). For this purpose, we performed an ANCOVA test, where the number of strategies was entered as the dependent variable, sex and relationship status were entered as the independent categorical variables, and age was entered as a continuous independent variable. No significant effects were found.

#### 3.2.4. Main Strategies

To classify the identified strategies into broader domains or main strategies, we conducted a second-order factor analysis. Specifically, we performed exploratory factor analysis using principal axis factoring for factor extraction and direct oblimin for rotation on the ten variables representing the mean scores of the identified factors. Parallel analysis indicated a three-domain solution, which is presented in Table 3.

The first main strategy to emerge was “Solve the problem”, where people aimed to address the problems that arose by making concessions and compromises, and by discussing and negotiating with their partners. Increasing physical contact and avoiding acting in the heat of the moment appeared to be facets of this strategy. The second main strategy to emerge was “External input”, where in order to manage relationship problems, individuals asked the opinion and advice of experts, friends, and family. Facets of this strategy included having a parallel relationship and doing things for themselves. The third main strategy to emerge was “Avoid the problem”, where people would avoid or ignore the problem. The “Spiritual coping” strategy also loaded here, but the factor loading was relatively low. Some cross-loadings were also identified. In particular, the “Parallel relationship” strategy also loaded onto the “Avoid the problem” (0.439) and the “Solve the problem” (−0.336) main strategies. In addition, the “Self-care actions” strategy also loaded onto the “Solve the problem” (0.321) main strategy. The most likely to be used main strategy was “Solve the problem” (*M* = 3.72, *SD* = 0.45), followed by “External input” (*M* = 2.53, *SD* = 0.56), and “Avoid the problem” (*M* = 2.11, *SD* = 0.70).

## 4. Discussion

In the present research, we employed a mixed-methods approach to identify the different strategies that people use to manage intimate relationship problems. Specifically, we identified 93 distinct actions that we classified into 10 broad strategies and 3 broader main strategies. The most likely to be used main strategy was “Solve the problem”, where participants attempted to solve the problem by discussing and negotiating it with their partners, making compromises, improving physical intimacy, and avoiding acting in the heat of the moment. The second most likely to be used main strategy was “External input”, where participants aimed to obtain advice and support from others, including friends, relatives, and psychologists. The least likely to be used strategy was “Avoid the problem”, where people attempted to manage their problems by ignoring or avoiding them. Participants indicated that they were likely to use a combination of strategies to manage their problems. Men and women were generally in agreement over which strategy to use, but age effects were found for several strategies. Next we will discuss the three main strategies in more detail.

In accordance with our original prediction and the findings of previous research ([5]), one main strategy to manage relationship problems is to attempt to resolve them when they arise. One way is for people to discuss the problems with their partners and attempt to negotiate a solution that would be beneficial to both. Negotiating a solution is unlikely to work unless the parties involved make some compromises, which is one facet of this strategy. Moreover, discussion and negotiations are unlikely to be effective if they take place in the heat of the moment, so another facet of this main strategy is to remain calm and avoid discussing the problems when the situation is heated. We also found that one facet of this main strategy was to attempt to become more intimate with one’s partner, seeking, for instance, more sexual contact. One reason could be that more intimacy could bring the parties involved closer and thus put them in a better position to negotiate solutions to their problems. Additionally, one common relationship problem is a lack of intimacy ([3]); thus, this sub-strategy can work in addressing it. Participants indicated “Solve the problem” to be the most likely to be used main strategy, pertaining to its effectiveness.

Also consistent with our original prediction, in their effort to manage relationship problems, people look for “External input”. In particular, they seek the advice of friends and relatives, as well as experts such as psychologists. People also observe and try to learn from relationships that work well and relationships that have failed. The sub-strategies “Parallel relationship” and “Self-care actions” also loaded here, although not very strongly. One possible reason for this involved turning to somebody or something outside the relationship to manage the issues that arise inside the relationship. Participants rated this as the second most likely strategy to adopt. To the best of our knowledge, this strategy has not been identified by research on conflict resolution, probably because it mainly aims to manage rather than solve relationship problems. That is, people can derive considerable emotional support and advice from discussing their problems with their friends and family or with experts, which in turn would enable them to better manage but not necessarily to solve the issues they face in their relationship.

As we originally predicted, one main strategy for managing the problems that arise in a relationship is to avoid or ignore them. The “Spiritual coping” sub-strategy also loaded here, albeit weakly. One reason could be that some participants interpreted turning to religion as a way to avoid dealing with relationship problems. “Avoid the problem” was indicated by participants as the least likely to be used main strategy.

Our prediction that people would engage in a parallel relationship to manage relationship problems was partially supported. More specifically, the “Parallel relationship” emerged as a strategy rather than a main strategy that was correlated with all the identified main strategies. Apart from the “External input”, it also loaded onto the “Avoid the problem” main strategy, one possible reason being that participants interpreted a parallel relationship as a way to avoid the problems people face in their legitimate relationship. It also loaded onto the “Solve the problem” main strategy, with the factor loading being negative. One interpretation of this finding is that having a parallel relationship not only cannot solve the existing problems people face in their legitimate relationship, but it can cause new ones.

We did not find significant sex differences in any of the identified strategies, indicating that men and women were generally in agreement over which strategies they were likely to use to manage relationship problems. In addition, no significant differences were found between participants who were in an intimate relationship and participants who were single. On the other hand, we found significant effects of age for most of the identified strategies. The largest effect was for the “Communication and problem-solving with partner” strategy, where older participants indicated a lower likelihood to use it than younger ones (for a discussion on the problems older couples face see ([22]; [24])). This effect could be due to changing patterns of communication with life stage. For instance, one study found that older couples were more likely to adopt less direct and less disclosing forms of communication during conflict than younger couples ([34]). The second largest effect was for the “Consult an expert” strategy, where older participants indicated a higher likelihood to use it than younger ones. One possible reason is that such consultation requires monetary resources that younger people are more likely to lack.

Participants indicated that they were likely to use more than one strategy; for instance, more than 27% indicated that they were likely to use five. This is reasonable for two main reasons. First, to manage a relationship problem, a combination of different strategies could be more effective than the use of a single one. For instance, as discussed above, a problem may not be resolved only by discussing it with a partner; it is more likely to be resolved if partners discuss it when they are calm and are likely to make compromises. The second reason is that people face a diversity of relationship problems ([3]), and different problems may require different ways to manage them. For example, a partner may have a character flaw, and because personality is relatively stable ([29]), it may work better for people to ignore it. Or the problem in the relationship may be a health issue that a partner faces, which people can better manage by deriving support from others such as a psychologist. Future research could attempt to identify which strategies people use to deal with specific problems.

One may wonder whether the strategies we have identified here are indeed effective in managing relationship problems. Making the reasonable assumption that most intimate relationships are not problem-free, and given that most people manage to keep one long enough to lead to marriage and children ([14]), we can conclude that these strategies are to some degree effective. On the other hand, the rising number of divorces ([30]) and rate of singlehood ([25]) suggest that the effectiveness of these strategies is also limited. One reason is the mismatch problem: the conditions under which people had to keep an intimate relationship in ancestral human societies were different from those in contemporary societies. Accordingly, these strategies have evolved to work best in the ancestral context, and they may fall short in enabling people to deal with relationship problems in contemporary societies. Future studies need to examine how effective the identified strategies are for managing relationship problems.

The effectiveness of a strategy or combination of strategies depends on the nature of the problem, as well as the individual characteristics of the party that uses it/them. As discussed previously, different problems may require different strategies to be effectively managed. Moreover, the traits of an individual would predict how effective each strategy would be. For instance, the “Communication and problem-solving with partner” strategy may be more effective when adopted by people with good communication skills than by people with poor ones. Accordingly, more research is required to identify how individual differences and the nature of the relationship problem predict the effectiveness of strategies used. This line of research could have practical value for people who are interested in managing their relationship problems, and for professionals such as counseling psychologists, who could use it to develop interventions that enable their clients to deal with the problems they face in their relationships more effectively. In particular, by identifying the different strategies that individuals can use and identifying which ones are more effective, a specialist could inform their clients about what can work for them.

One limitation of the current research is that it was based on self-report data, which are subject to several biases, including participants giving inaccurate answers. Additionally, our study is cross-sectional; thus, significant results should not be interpreted as causal relationships. Moreover, we employed non-probability samples, so our findings may not readily generalize to the population (but see [15]). Additionally, our study was confined to the Greek cultural context, so its findings may not readily generalize to different cultural settings. One reason can be that cultural factors may affect the likelihood of using some of the identified strategies. It could be the case, for instance, that in some cultures, it is more acceptable to discuss relationship problems with others, such as a psychologist, than in others. Thus, future studies need to take a cross-cultural perspective, attempting to identify how cultural factors can affect the likelihood of people using the identified strategies. Furthermore, there is likely a plethora of factors, such as personality traits, that predict one’s likelihood to use the identified strategies. In the present study, we have only examined the effects of sex and age, and future studies need to expand our work by examining the effects of additional factors.

Intimate relationships are not problem-free, and people who want to keep one in the long term need to find some way to manage the problems that arise in it. In the current research, we have identified three main strategies consisting of ten strategies that people use for this purpose. More work is required, however, to identify the effectiveness of each of the identified strategies.

## Figures and Tables

**Figure 1 behavsci-15-00218-f001:**
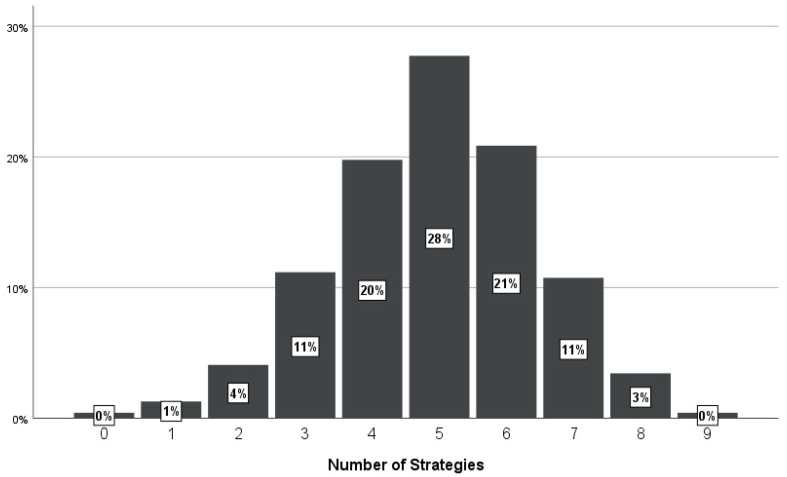
The figure above depicts how many of the 10 identified strategies participants indicated that they were highly likely to use.

**Table 1 behavsci-15-00218-t001:** The strategies for managing relationship problems identified in Study 1 and their classification into broader categories in Study 2.

Strategies/Acts	Factor Loadings	Cronbach’s α
Communication and problem-solving with partner		0.91
I discuss the problems with my partner	0.816	
I talk openly with my partner about how I feel	0.721	
I ask my partner how he/she thinks we can deal with our problems	0.674	
I explain my concerns to my partner with arguments	0.660	
I try to identify the cause of the problem	0.622	
I try to address the cause of the problem	0.618	
I try to improve communication with my partner	0.587	
I try to understand my partner’s concerns	0.585	
I seek to find common ground with my partner	0.557	
I try to say that it bothers me	0.556	
I am honest with my partner about my concerns	0.541	
I try to listen attentively to my partner’s concerns	0.533	
I show my partner that I understand his/her concerns	0.500	
I try to see my partner’s side	0.485	
I try to be realistic about the relationship and its problems	0.441	
I try to figure out if I am doing something wrong and fix it	0.426	
I try to recognize my own weaknesses	0.418	
I don’t tell him/her things that bother me	−0.416	
I make efforts to improve myself	0.410	
I show empathy	0.402	
I seek to be open to changes that can help the relationship	0.392	
I try to recognize that it is not possible to agree on everything	0.365	
External input and other relationships observation		0.84
I discuss problems with others who are in a long-term relationship	0.793	
I discuss the problems with my friends/acquaintances	0.784	
I ask for advice from familiar persons	0.784	
I observe others in relationship and how they deal with problems	0.685	
I observe other relationships that seem to work well	0.631	
I discuss problems with my relatives	0.544	
I consider other relationships that have broken up and try to figure out what went wrong	0.516	
I watch videos/podcasts on this topic	0.387	
Concessions and compromises		0.79
I make concessions	0.697	
I make compromises	0.692	
I apologize even if it’s not my fault	0.590	
I take a step back	0.542	
I am patient	0.524	
I try to be more tolerant towards my partner	0.520	
I forgive my partner’s behaviors	0.515	
I apologize if I am at fault	0.432	
I justify my partner’s behaviors	0.420	
I take responsibility for things I didn’t do well	0.414	
I avoid criticizing my partner	0.301	
Reflective response		0.87
I avoid reacting directly but take time to think first	0.754	
I avoid talking when I am angry	0.649	
I don’t make decision on the heat of the moment	0.639	
I don’t make hasty decisions	0.583	
I try to stay calm	0.541	
I try to keep calm	0.518	
I avoid reacting	0.485	
I try to distance myself from the problem so that I can see it better	0.457	
I give space and time to my partner after an argument	0.441	
Consult an expert		0.87
I discuss the problems with a psychologist	−0.907	
I seek help from a psychologist/psychiatrist	−0.906	
I do psychotherapy	−0.888	
I try to begin psychological counseling with my partner	−0.702	
I read up on psychology	−0.638	
I read articles/books that refer to the topic	−0.489	
I taking a psychoactive drug (e.g., sedative)	−0.349	
Intimacy enhancement efforts		0.78
I try to have more sex with my partner	0.736	
I try to have more physical contact with my partner	0.735	
I seek to spend more quality time with my partner	0.449	
I make the relationship my priority	0.368	
I try to do more activities with my partner that we both enjoy	0.365	
Spiritual coping		0.81
I turn to religion	−0.891	
I pray to God to solve the problems	−0.866	
I discuss the problems with a priest/spiritual advisor	−0.727	
Parallel relationship		0.66
I have a relationship on the side	0.707	
I live a parallel life	0.688	
I distance myself from the relationship for a while	0.465	
I spend a short time away from my partner	0.435	
I keep myself busy with work	0.417	
Self-care actions		0.76
I do things for myself	0.757	
I do physical exercise/activities	0.658	
I try to do things that please me	0.649	
I take time for myself to improve	0.576	
I focus on myself	0.555	
I meditate	0.362	
I spend time with myself to think about the problem	0.325	
I try not to hold myself back	0.317	
Avoidance		0.76
I try not to think about the problem	0.672	
I try not to think about the problem for a while	0.651	
I’m downplaying the problem	0.617	
I try to distance myself from the problem	0.561	
I ignore the problem hoping it will go away	0.510	
I ignore the problem	0.502	

**Table 2 behavsci-15-00218-t002:** Mean scores, sex, age, and relationship status effect in Study 2.

Strategies		Overall	Women	Men	Sex		Age		Relationship Status	
	%	Mean (SD)	Mean (SD)	Mean (SD)	*p*-Value	η_p_^2^	*p*-Value	η_p_^2^	*p*-Value	η_p_^2^
Communication and problem-solving with partner	96.5	4.04 (0.52)	4.08 (0.51)	3.99 (0.52)	0.312	0.050	(−) <0.001	0.154	0.607	0.043
Intimacy enhancement efforts	81.3	3.77 (0.77)	3.77 (0.80)	3.76 (0.74)	0.052	0.018	(−) <0.001	0.077	0.239	0.013
Concessions and compromises	84.5	3.61 (0.58)	3.58 (0.59)	3.63 (0.56)	0.430	0.023	(−) <0.001	0.070	0.330	0.025
Reflective response	72.3	3.46 (0.65)	3.41 (0.70)	3.52 (0.60)	0.017	0.034	0.132	0.028	0.225	0.023
Self-care actions	64.4	3.30 (0.68)	3.29 (0.68)	3.30 (0.68)	0.876	0.010	(−) <0.001	0.062	0.816	0.011
External input and other relationships observation	35.6	2.70 (0.92)	2.78 (0.91)	2.60 (0.92)	0.100	0.024	(−) <0.001	0.087	0.806	0.011
Avoidance	14.0	2.30 (0.73)	2.42 (1.06)	2.38 (0.75)	0.486	0.012	0.587	0.010	0.906	0.006
Consult an expert	23.8	2.19 (1.03)	2.23 (0.71)	1.94 (0.73)	0.009	0.030	(+) <0.001	0.098	0.827	0.009
Parallel relationship	5.4	1.93 (0.70)	2.06 (1.21)	1.91 (0.93)	0.120	0.016	(+) <0.001	0.059	0.025	0.021
Spiritual coping	15.5	1.92 (1.12)	1.91 (0.67)	1.77 (1.00)	0.948	0.002	(+) <0.001	0.060	0.707	0.004

Note: The signs in parenthesis indicate the direction of the relationship. Note 2: The “%” column reports the percentage of participants who had a mean score above “3” in each factor. Note 3: The significance level has been set to 0.005.

**Table 3 behavsci-15-00218-t003:** The extracted main strategies in Study 2.

Main Strategy	Factor Loadings
Solve the problem	
Concessions and compromises	0.787
Communication and problem-solving with partner	0.752
Intimacy enhancement efforts	0.649
Reflective response	0.604
External input	
Consult an expert	0.752
External input and other relationships observation	0.720
Parallel relationship	0.587
Self-care actions	0.547
Avoid the problem	
Avoidance	0.845
Spiritual coping	0.382

## Data Availability

All data and materials are available here: https://osf.io/ygr8e/?view_only=c3efe688e7ef4bed84cf0202c2bec732 (accessed on 20 October 2024).

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
