# Peer review of "How People Manage Relationship Problems: An Exploratory Study in the Greek Cultural Context"

_behavsci, 2025, doi:10.3390/bs15020218_

Round 1
Reviewer 1 Report
Comments and Suggestions for Authors
Behavioral sciences
Manuscript ID behavsci-3379381
How people manage relationship problems: An exploratory study in the Greek cultural context
The manuscript is certainly an interesting read, delving into a relevant topic with a well-performed mixed-methods approach. The writing and structure are of good quality, and the link between theory, the qualitative results and the quantitative results can be highly useful for both further research and practical applications. However, several limitations and unclarities should still be addressed in order to make the paper truly fit for publication. These are listed below in chronological order, and concern both minor and more major suggestions/issues.
Abstract
R.19) “and classified these actions into 10 broad strategies and three broader main strategies”: this is quite vague and incomprehensive. Considering the rest of the abstract only goes into the three broader strategies, it seems relevant only mention those. An alternative is to use different concepts to clarify the difference in the abstract. For example, the title of Table 2 refers to “domains” in reference to 3 main strategies.
Introduction
R.62-63) “For example, people vary in personality traits (Matthews et al., 2015), and intimate partners may differ in these dimensions”: this phrasing is somewhat excessive. Following the previous sentence, something like “Intimate partners may, for instance, differ in terms of personality traits”.
R.65-66) “The mismatch problem may be another source of relationship problems.”: this sounds very strange to me. Wasn’t the problem of mismatched personality traits just identified? How this links to the comparison of the ancestral versus modern environment is also unclear.
R.66-R.79) This entire section seems odd here. The added value of comparing ancestral and modern environments is somewhat lost on me in illustrating the nature of relationship problems (i.e., the title of this section). Furthermore, “it follows that adaptations involved in mating may have been shaped by election forces to motivate an aggressive response even for trivial disagreements” seems to underline that aggressive responses were evolutionarily developed as beneficial to mating, while sociobiological theory rather underlines the (biological) “feminization” of males and the importance of being calm and reserved, rather than aggressive, as females more likely look for a long-term partner than a mere “fertilizer”. But the text then goes on to say “Such responses could be problematic in contemporary relationships, with their root being the mismatch problem rather than conflict between the sexes”. I’m thus a bit lost as to where the authors are headed with this section.
R.82) “To begin with, one study employed”: but then two studies are cited, of two very different time periods.
R.86) “Another study examined”: but the same source is used as in the previous example.
Section 1.2) This section reads as a mere recital of random studies, and lacks integration. Infidelity, for example, is mentioned several times separately, while it would seem more logical to stress that this is a persisting problem for relationships, across time and space. As the section is titled “Problems in Intimate Relationships”, I would like to see an integrated overview of the most notable problems that span culture and time periods.
R.100-102) “Forming long-term intimate relationships is not a recent phenomenon, as there are reasons to believe that across different cultures and times, people have formed such relationships (Fisher, 2017)”: It seems a bit strange to formulate this so suggestively and vaguely. I would suggest presenting this in a more factual manner, as the next sentence (which also states that “there are also reasons (…) to believe”) then follows more smoothly.
R.107) “We will discuss such strategies next.”: This can be deleted, in my opinion.
R.113-115): “For such negotiations to succeed, some compromise may be necessary. Research on conflict resolution indicates that making compromises is a common strategy (Kurdek, 1994; Zacchilli et al., 2009).”: This could be reworked into one sentence as to avoid repetition of the word “compromise”
R.116) “The second strategy we propose”: This phrasing seems out of place in the literature overview. However, the lack of sources here (R.119-126) does seem to indicate that this is “advice” by the authors, which is particularly strange. Moreover, the next paragraph again refers to research on conflict resolution. I would suggest reworking the entire section 1.3 into presenting the theory on conflict resolution in light of relationship problems. Currently, the section seems like a pick-and-choose by the authors, rather than a comprehensive overview of the literature pertaining to relationship management. Moreover, this seemingly perfectly matches with the results, which is odd. Especially in light of the qualitative research goals and the exploratory nature of the research, a broader overview is in place here.
Section 1.4) As the previous section does not grant a sufficient overview of existing research into conflict resolution, it remains somewhat unclear how the focus of the current study expands upon and contributes to the research field. Much has already been said concerning relationship management, which is important to clearly address in order to bring across the added value of the current study. Moreover, the differentiation between conflict and problem is currently not very convincing, as the authors already discussed the tactic of conflict avoidance as being effective when “when the problem causes little disturbance to the relationship and/or cannot be easily resolved”. Finally, the examination of sex, age and relationship status differences now seems rather ad-hoc, as this is not addressed in the literature overview (nor discussed in-depth in the discussion section).
R.157) “To the best of our knowledge, this is the first study in the area”: This is, again, not very convincing, as plenty of related research was just mentioned. It seems crucial for the authors to pinpoint the exact added value of the current study vis-à-vis the plethora of work already done on relationship management and conflict resolution.
Study 1-2
Table 1) Why is “I don't tell him/her things that bother me” under Communication and problem-solving, rather than avoidance? I understand that, content-wise, it matches with the communication factor, but I would like some reflection on the potential overlap of the items.
Table 1) Typo: I watch videos/podcasts that on this topic
Table 1) The phrasing differs between a general statement (e.g. “I try”) and an expression of current actions (“I am trying”). This reads somewhat strangely, as later on in the results section, these expressions are used to gauge the likelihood of using a certain strategy. From the likelihood perspective, “I try to stay calm” and “I am trying to keep calm” should not be considered as separate actions, but as they are listed as such, I am confused as to how they relate differently to the results.
R.235) “Moreover, we conducted a series of MANCOVA tests to identify significant effects.”: I would like some more clarification here of the precise goal. This becomes somewhat clear in the next sentences (i.e. assessing differences by gender, relationship status and age), but then the phrasing of “effects” is incorrect. Rather, “associations” would be better, as well as the immediate emphasis of the demographics used for the comparison.
R.236-238) “Specifically, for each identified strategy, the actions within that strategy entered as dependent variables. Sex and relationship status entered as the independent categorical variables, while age entered as a continuous independent variable.”: I think the passive form should be used (“were”/”was” entered). Furthermore, next sentence (“This analysis was performed for each strategy”) is repetitive and can be removed.
Results
R.247) “In total, nine items were removed, and they are listed in the Appendix.”: I would suggest rephrasing this as “.. were removed, which are listed in the Appendix”.
R.264-265) To manage relationship problems, people would also “Consult an expert” such as a psychologist, and they may also engage in couples therapy.: There is seemingly a switch here between the past and present tense. More concise would be “.. such as a psychologist, or engage in couples therapy”.
R.276) “Likelihood of Use” is the title of the section, but what is described is the willingness to use the strategies. While these are certainly related, the sentiment is still different. I would therefore suggest that the authors refrain from using the “willingness” phrasing, as this is not what is measured. This also ties into my comment on the phrasing of items in Table 1 in terms of general statements (e.g. “I try”) versus current actions (e.g. “I am trying”). Whereas someone may not be willing to have a parallel relationship, this may nevertheless be the current scenario (and thus seen as more likely in the future as well). The reflection on likelihood versus willingness in light of what people are currently doing versus what they want to be doing is now lacking in the paper.
In general, I would suggest refraining from referring to “effects”. Tested here are associations or differences.
Figure 1 is missing from the manuscript and appendix
Table 2) As the significance level is set at .005, it would be useful to see this reflected in the table itself – or at least repeated in the notes. Maybe also put the significant associations in bold print in the table?
Discussion
R.354-355) “we classified into ten broad strategies and three broader main strategies”: again, the phrasing could maybe be changed here (e.g. to dimensions, as is the title of Table 2).
R.364) Again, I would suggest refraining from using the term “effect”, and rather use “association” or something less causal.
R.365) “In accordance with our original prediction”: This is strange to read, as it was previously explicitly stated that no hypotheses were formulated due to this being exploratory research. Moreover, this highlights the issue designated in an earlier comment: that the literature overview seemingly seamlessly matches with the results. Especially in light of the qualitative research focus, this feels forced and too limited. The same concerns R.380, R.393 and R.398.
R.378) “Participants indicated the “Solve the problem” to be the most likely”: Typo, the first “the” can be removed.
R.368-385) “… also loaded here, albeit not very strongly, one possible reason being that…”: This sentence is incorrectly structured.
R.413-R.415) “The largest effect was for the “Communication and problem-solving with partner” strategy, where older participants indicated a lower willingness to use it than younger ones. We do not have a working hypothesis explaining this effect.”: This seems odd, as I would certainly expect relationship maintenance is certainly to differ – for example older cohorts being less adept at communicating than younger ones.
R.433-435) “Making the reasonable assumption that … we can reasonably conclude”: Repeated use of “reasonable” in the same sentence.
R.440-441) “Thus” and “so” are both explanatory/conclusive terms, which is strange in the same sentence. Furthermore, just as in the literature overview, it is unclear to me what the authors mean by “selection forces” when comparing the ancestral to the modern context.
Author Response
Reviewer 1
The manuscript is certainly an interesting read, delving into a relevant topic with a well-performed mixed-methods approach. The writing and structure are of good quality, and the link between theory, the qualitative results and the quantitative results can be highly useful for both further research and practical applications. However, several limitations and unclarities should still be addressed in order to make the paper truly fit for publication. These are listed below in chronological order, and concern both minor and more major suggestions/issues.
We would like to thank you very much for considering our paper and for your kind words about our work. We truly appreciate your effort in providing us detailed feedback that enabled us to improve the manuscript. Please see below how we have addressed all your concerns and recommendations.
We would like to take this opportunity to say that, although most people would at some point find themselves in the scenario where they have to manage relationship problems, there is little if any research on how they do so. We believe that, despite its limitations, the research presented in the current paper has covered considerable distance, and we hope that it will trigger further research that is necessary for understanding this aspect of human mating.
Abstract
R.19) “and classified these actions into 10 broad strategies and three broader main strategies”: this is quite vague and incomprehensive. Considering the rest of the abstract only goes into the three broader strategies, it seems relevant only mention those. An alternative is to use different concepts to clarify the difference in the abstract. For example, the title of Table 2 refers to “domains” in reference to 3 main strategies.
Yes, following your suggestion we now refer only to the three main strategies. Also, for consistency we dropped the “domain” from the title of Table 3.
Introduction
R.62-63) “For example, people vary in personality traits (Matthews et al., 2015), and intimate partners may differ in these dimensions”: this phrasing is somewhat excessive. Following the previous sentence, something like “Intimate partners may, for instance, differ in terms of personality traits”.
Good point, we have changed this accordingly (61).
R.65-66) “The mismatch problem may be another source of relationship problems.”: this sounds very strange to me. Wasn’t the problem of mismatched personality traits just identified? How this links to the comparison of the ancestral versus modern environment is also unclear.
Good point. With the term “mismatch” we refer to the evolutionary mismatch between ancestral and contemporary conditions. Accordingly in the present revision, we now state ‘evolutionary mismatch’ (64)
R.66-R.79) This entire section seems odd here. The added value of comparing ancestral and modern environments is somewhat lost on me in illustrating the nature of relationship problems (i.e., the title of this section). Furthermore, “it follows that adaptations involved in mating may have been shaped by election forces to motivate an aggressive response even for trivial disagreements” seems to underline that aggressive responses were evolutionarily developed as beneficial to mating, while sociobiological theory rather underlines the (biological) “feminization” of males and the importance of being calm and reserved, rather than aggressive, as females more likely look for a long-term partner than a mere “fertilizer”. But the text then goes on to say “Such responses could be problematic in contemporary relationships, with their root being the mismatch problem rather than conflict between the sexes”. I’m thus a bit lost as to where the authors are headed with this section.
In this section, we aim to understand the ultimate reasons behind relationship problems. We have argued that one such reason is conflict between the sexes, while another reason is the mismatch problem. For example, there are reasons to believe that most of human evolution took place in a context where human rights were not well protected, and men dominated over women. Thus, it could be the case that men, who are on average bigger and stronger than women, have evolved a disposition to employ physical aggression to impose their will on their wives. In a contemporary Western context, where human rights are well protected, and women are not forced in relationships, such male disposition would be problematic, leading to relationship strain. In the current revision, we have reworked this section to make the argument clearer to the reader (71-79).
R.82) “To begin with, one study employed”: but then two studies are cited, of two very different time periods.
You are right, only Bloom et al., 1985 needs to be cited here- this error has been corrected (86).
R.86) “Another study examined”: but the same source is used as in the previous example.
Section 1.2) This section reads as a mere recital of random studies, and lacks integration. Infidelity, for example, is mentioned several times separately, while it would seem more logical to stress that this is a persisting problem for relationships, across time and space. As the section is titled “Problems in Intimate Relationships”, I would like to see an integrated overview of the most notable problems that span culture and time periods.
Your point is sound, we regret to say however, that to the best of our knowledge, there has not been any study that has provided “an integrated overview of the most notable problems that span culture and time periods.” In the current section, we discussed problems that have been identified across different studies, but we agree that more studies are need to provide what you ask. We think that it is big gap in the literature, and we hope that it will be addressed in the future.
R.100-102) “Forming long-term intimate relationships is not a recent phenomenon, as there are reasons to believe that across different cultures and times, people have formed such relationships (Fisher, 2017)”: It seems a bit strange to formulate this so suggestively and vaguely. I would suggest presenting this in a more factual manner, as the next sentence (which also states that “there are also reasons (…) to believe”) then follows more smoothly.
Following your suggestion, we have added one more sentence to clarify our argument (103).
R.107) “We will discuss such strategies next.”: This can be deleted, in my opinion.
Done! (110)
R.113-115): “For such negotiations to succeed, some compromise may be necessary. Research on conflict resolution indicates that making compromises is a common strategy (Kurdek, 1994; Zacchilli et al., 2009).”: This could be reworked into one sentence as to avoid repetition of the word “compromise”
This is a good idea, we have rephrased the sentence accordingly (116).
R.116) “The second strategy we propose”: This phrasing seems out of place in the literature overview. However, the lack of sources here (R.119-126) does seem to indicate that this is “advice” by the authors, which is particularly strange. Moreover, the next paragraph again refers to research on conflict resolution. I would suggest reworking the entire section 1.3 into presenting the theory on conflict resolution in light of relationship problems. Currently, the section seems like a pick-and-choose by the authors, rather than a comprehensive overview of the literature pertaining to relationship management. Moreover, this seemingly perfectly matches with the results, which is odd. Especially in light of the qualitative research goals and the exploratory nature of the research, a broader overview is in place here.
It is important to say here that the present study is not about conflict resolution but about managing relationship problems. This is a crucial distinction, because managing relationship problems is not the same as resolving them – many if not most of the couples do not completely solve their problems but find ways to live with them – identifying these ways is the purpose of the current work. We have expanded and reworked the first paragraph of the 1.4 section to make this argument more explicit. In this respect, it would not be appropriate to expand on conflict resolution as a theoretical framework for the current work.
Section 1.4) As the previous section does not grant a sufficient overview of existing research into conflict resolution, it remains somewhat unclear how the focus of the current study expands upon and contributes to the research field. Much has already been said concerning relationship management, which is important to clearly address in order to bring across the added value of the current study. Moreover, the differentiation between conflict and problem is currently not very convincing, as the authors already discussed the tactic of conflict avoidance as being effective when “when the problem causes little disturbance to the relationship and/or cannot be easily resolved”. Finally, the examination of sex, age and relationship status differences now seems rather ad-hoc, as this is not addressed in the literature overview (nor discussed in-depth in the discussion section).
R.157) “To the best of our knowledge, this is the first study in the area”: This is, again, not very convincing, as plenty of related research was just mentioned. It seems crucial for the authors to pinpoint the exact added value of the current study vis-à-vis the plethora of work already done on relationship management and conflict resolution.
As stated above, we have reworked this section to make a better distinction between conflict resolution and managing relationship problems. Resolving problems is only one strategy for managing them. This argument is also consistent with our findings, as we have identified many strategies that clearly do not aim to resolve problems (e.g., Spiritual coping, Parallel relationship etc.). To best of our knowledge, there is no study that has attempted to identify these strategies, which we consider an important omission in the literature that the current study aims to address. This omission most probably comes from people viewing resolving relationship problems to equal managing relationship problem, but a more accurate way – consistent with the presented empirical evidence- is that resolving relationship problems is one of the ways to manage them. Moreover, you right about the sex, age, and relationship status difference – we did not discuss them in the theoretical part because we did not have a priori hypotheses about how they would predict the identified strategies. Yet, given the nature of the question we aim to address, we believe that it is important to examine these factors.
Study 1-2
Table 1) Why is “I don't tell him/her things that bother me” under Communication and problem-solving, rather than avoidance? I understand that, content-wise, it matches with the communication factor, but I would like some reflection on the potential overlap of the items.
One possible explanation is that participants see this more as a communication with their partner than avoiding him/her. Please note that this classification was not made by us, but by the analysis performed (i.e., principal axis factoring).
Table 1) Typo: I watch videos/podcasts that on this topic
Thanks! This has been corrected.
Table 1) The phrasing differs between a general statement (e.g. “I try”) and an expression of current actions (“I am trying”). This reads somewhat strangely, as later on in the results section, these expressions are used to gauge the likelihood of using a certain strategy. From the likelihood perspective, “I try to stay calm” and “I am trying to keep calm” should not be considered as separate actions, but as they are listed as such, I am confused as to how they relate differently to the results.
Thank you for spotting this issue. The correct phrasing is “I try” rather than “I am trying.” This error came from translating the items from Greek to English, as in Greek there is no distinction between the two. This has been corrected throughout the manuscript. Please not that we did so for other items as well.
R.235) “Moreover, we conducted a series of MANCOVA tests to identify significant effects.”: I would like some more clarification here of the precise goal. This becomes somewhat clear in the next sentences (i.e. assessing differences by gender, relationship status and age), but then the phrasing of “effects” is incorrect. Rather, “associations” would be better, as well as the immediate emphasis of the demographics used for the comparison.
Yes, following your suggestion we have replaced “effects” with “associations.” (251)
R.236-238) “Specifically, for each identified strategy, the actions within that strategy entered as dependent variables. Sex and relationship status entered as the independent categorical variables, while age entered as a continuous independent variable.”: I think the passive form should be used (“were”/”was” entered). Furthermore, next sentence (“This analysis was performed for each strategy”) is repetitive and can be removed.
Following your suggestion the passive form is now used (252). Moreover, the sentence was deleted (254).
Results
R.247) “In total, nine items were removed, and they are listed in the Appendix.”: I would suggest rephrasing this as “.. were removed, which are listed in the Appendix”.
Done! (262)
R.264-265) To manage relationship problems, people would also “Consult an expert” such as a psychologist, and they may also engage in couples therapy.: There is seemingly a switch here between the past and present tense. More concise would be “.. such as a psychologist, or engage in couples therapy”.
We have changed this sentence in accordance with your suggestion (279).
R.276) “Likelihood of Use” is the title of the section, but what is described is the willingness to use the strategies. While these are certainly related, the sentiment is still different. I would therefore suggest that the authors refrain from using the “willingness” phrasing, as this is not what is measured. This also ties into my comment on the phrasing of items in Table 1 in terms of general statements (e.g. “I try”) versus current actions (e.g. “I am trying”). Whereas someone may not be willing to have a parallel relationship, this may nevertheless be the current scenario (and thus seen as more likely in the future as well). The reflection on likelihood versus willingness in light of what people are currently doing versus what they want to be doing is now lacking in the paper.
Thank you for this, it is a good point. Accordingly, in this revision we have dropped the “willingness” term and we use “likelihood” as it more accurately reflects our results.
In general, I would suggest refraining from referring to “effects”. Tested here are associations or differences.
We see your point. However, as we have entered, sex, age, and relationship status as independent variables, the usual practice is to call them effects.
Figure 1 is missing from the manuscript and appendix
Yes, this was an error, the figure is now added in the text.
Table 2) As the significance level is set at .005, it would be useful to see this reflected in the table itself – or at least repeated in the notes. Maybe also put the significant associations in bold print in the table?
Yes, following your suggestion we have added a note below table 2 to indicate that the significance level was set to .005.
Discussion
R.354-355) “we classified into ten broad strategies and three broader main strategies”: again, the phrasing could maybe be changed here (e.g. to dimensions, as is the title of Table 2).
Given the topic of the study, we believe that “strategies” is a more accurate term than “dimensions.”
R.364) Again, I would suggest refraining from using the term “effect”, and rather use “association” or something less causal.
Following your suggestion, in the current revision, we state explicitly that our study is cross-sectional so, our results should not be interpreted as causal relationships (510).
R.365) “In accordance with our original prediction”: This is strange to read, as it was previously explicitly stated that no hypotheses were formulated due to this being exploratory research. Moreover, this highlights the issue designated in an earlier comment: that the literature overview seemingly seamlessly matches with the results. Especially in light of the qualitative research focus, this feels forced and too limited. The same concerns R.380, R.393 and R.398.
Our argument, as stated in the introduction, is that we can draw a number of hypotheses about the strategies people are likely to use in order to manage relationship problems, yet because the phenomenon is complex and there is limited research in the area, we cannot predict all these strategies, so our study was exploratory. Moreover, we have hypothesised the presence of four strategies, and in the results indicated the presence of three main strategies and 10 strategies.
R.378) “Participants indicated the “Solve the problem” to be the most likely”: Typo, the first “the” can be removed.
Done! (435)
R.368-385) “… also loaded here, albeit not very strongly, one possible reason being that…”: This sentence is incorrectly structured.
We have reworked and divided this sentence into two (440).
R.413-R.415) “The largest effect was for the “Communication and problem-solving with partner” strategy, where older participants indicated a lower willingness to use it than younger ones. We do not have a working hypothesis explaining this effect.”: This seems odd, as I would certainly expect relationship maintenance is certainly to differ – for example older cohorts being less adept at communicating than younger ones.
Yes, we also expect age differences, yet it is not very clear why older people would be less adept at communication than younger ones. It could be argued that older people are more experienced and thus, are better in communicating with their partners.
R.433-435) “Making the reasonable assumption that … we can reasonably conclude”: Repeated use of “reasonable” in the same sentence.
Following your suggestion the second “reasonable” has been deleted (488).
R.440-441) “Thus” and “so” are both explanatory/conclusive terms, which is strange in the same sentence. Furthermore, just as in the literature overview, it is unclear to me what the authors mean by “selection forces” when comparing the ancestral to the modern context.
We have rephrased the sentence dropping also the “selection forces” term (492).
We would like to thank you very much once more for providing us with a thorough review!

Reviewer 2 Report
Comments and Suggestions for Authors
This is an empirical, exploratory article, as stated by the authors, which aims to address the issue of how people manage relationship problems in intimate relationships. Although it is not a new topic, the article makes a more comprehensive analysis of the study of the various strategies available for this management, opting for a mixed methodology, qualitative and quantitative, which adds interest and scientific contribution to the research. Although the summary summarises the overall results, it can be improved by including what will be highlighted and how at the end of this work.
This discussion is well-founded and theoretically supported by various relevant empirical studies. However, it would be important to supplement many of the statements with more current studies from the last five years. The article is up-to-date and relevant, but it should consider the state of the art on the phenomenon over time. It is therefore suggested that the paragraph following the introduction should only appear after the conceptual and empirical review, in the section before the empirical parts.
In terms of method, no questions were raised in either study. In both studies, the sample, materials and procedures used were objectively characterised, including (at the end) mention of the ethics committee's decision code and the analyses carried out. The statistical analysis strategies have been adequately explained and are consistent with the objectives set. The results have been presented in an open and clear manner, discussing, taking into account all outupts and the literature on the subject. The authors discuss possible explanations, present the limitations of the study and the practical implications of the work.
In formal terms there is uniformity in the use of an editing standard, the authors referenced in the text are listed (and vice versa) and the length of the article is adequate.
Author Response
Reviewer 2
We would like to thank you very much for considering our paper and for your kind words about our work. We would like to take this opportunity to say that although the majority of people would find themselves in a scenario where they have to manage relationship problems, there has been limited if any research on how they do so. We believe that the current study constitutes an important first step in exploring this dimension of human mating.
This is an empirical, exploratory article, as stated by the authors, which aims to address the issue of how people manage relationship problems in intimate relationships. Although it is not a new topic, the article makes a more comprehensive analysis of the study of the various strategies available for this management, opting for a mixed methodology, qualitative and quantitative, which adds interest and scientific contribution to the research. Although the summary summarises the overall results, it can be improved by including what will be highlighted and how at the end of this work.
The focus of the Discussion section was to discuss the three main strategies. Following your suggestion, we have added a sentence to discuss it (423).
This discussion is well-founded and theoretically supported by various relevant empirical studies. However, it would be important to supplement many of the statements with more current studies from the last five years. The article is up-to-date and relevant, but it should consider the state of the art on the phenomenon over time. It is therefore suggested that the paragraph following the introduction should only appear after the conceptual and empirical review, in the section before the empirical parts.
In the introduction section, we attempted to provide a balance of older and more recent references. Given the limited length that a research paper should be, we think that we have covered adequately the existing literature, but if you have any additional study in mind please let us know and we will be happy to add it in the text. We would also like to note that, to the best of our knowledge, there is no other study to reference that has attempted to examine the different strategies that people use to manage relationship problems.
To be honest, we are not sure about which paragraph you are referring. Could you please specify?
In terms of method, no questions were raised in either study. In both studies, the sample, materials and procedures used were objectively characterised, including (at the end) mention of the ethics committee's decision code and the analyses carried out. The statistical analysis strategies have been adequately explained and are consistent with the objectives set. The results have been presented in an open and clear manner, discussing, taking into account all outupts and the literature on the subject. The authors discuss possible explanations, present the limitations of the study and the practical implications of the work.
In formal terms there is uniformity in the use of an editing standard, the authors referenced in the text are listed (and vice versa) and the length of the article is adequate.
Thank you very much once more for your feedback!

Round 2
Reviewer 1 Report
Comments and Suggestions for Authors
I thank the authors for the revision. Most of my points of concern have been sufficiently addressed, and I am now better convinced of the originality and contribution to scholarship of the work. Four final points remain, however, that I believe warrant more consideration before the manuscript is fit for publication.
Section 1.3) I had commented that this section i) “reads like a pick-and-choose by the authors, rather than a comprehensive overview of the literature pertaining to relationship management” and that ii) “this seemingly perfectly matches with the results, which is odd - especially in light of the qualitative research goals and the exploratory nature of the research”. In response, the authors state that “the present study is not about conflict resolution but about managing relationship problems” and “it would not be appropriate to expand on conflict resolution as a theoretical framework”. This, however, does not answer my comments. Why are strategies “proposed” – which match seemingly perfectly with the results? My point is not that the study is seemingly about conflict resolution, but that, the novelty of the work (in light of existing studies) could be made much clearer upfront. The addition to section 1.4 is beneficial in this respect, but would be more convincing if it was stated at the beginning of section 1.3.
A similar issue pertains to the discussion section. On the one hand, the authors state that no hypotheses were formulated due to this being exploratory research. However, in their response letter, the authors state “Our argument, as stated in the introduction, is that we can draw a number of hypotheses about the strategies people are likely to use in order to manage relationship problems, yet because the phenomenon is complex and there is limited research in the area, we cannot predict all these strategies, so our study was exploratory”. Why remain so intent on the use of hypotheses? Deductive research builds on existing theory and from that, deduces hypotheses. Inductive research, does the opposite: explorative work to produce theories. Considering that the authors also answer “To best of our knowledge, there is no study that has attempted to identify these strategies, which we consider an important omission in the literature that the current study aims to address”, it simply feels forced to want to portray this work as both exploratory and responsive to hypotheses. I understand your aim, but I suggest adhering completely to the exploratory strength of the paper.
It was found that older participants were less likely to use “communication and problem solving” as a strategy than younger participants. The authors answer that “it is not very clear why older people would be less adept at communication than younger ones. It could be argued that older people are more experienced and thus, are better in communicating with their partners”. I still expect reflection on this in the discussion. Emotional avoidance, the perception of issues and established (non-)communicative patterns in this respect are all valid and well-discussed explanations that can be found in the literature. Some examples:
- Zietlow, P. H., & Sillars, A. L. (1988). Life-stage differences in communication during marital conflicts. Journal of Social and Personal Relationships, 5(2), 223-245.
- Henry, R. G., Miller, R. B., & Giarrusso, R. (2005). Difficulties, disagreements, and disappointments in late-life marriages. The International Journal of Aging and Human Development, 61(3), 243-264.
- Jensen, J. F., & Rauer, A. J. (2015). Marriage work in older couples: Disclosure of marital problems to spouses and friends over time. Journal of Family Psychology, 29(5), 732.
Figure 1 was added to the manuscript, but the caption refers to “willingness” to use strategies – which, throughout the rest of the manuscript, the authors have changed to “likelihood”.
Author Response
I thank the authors for the revision. Most of my points of concern have been sufficiently addressed, and I am now better convinced of the originality and contribution to scholarship of the work. Four final points remain, however, that I believe warrant more consideration before the manuscript is fit for publication.
Thank you very much once more for considering our work and for providing us with additional feedback. Please see below how we have addressed all your remaining concerns and recommendations.
Section 1.3) I had commented that this section i) “reads like a pick-and-choose by the authors, rather than a comprehensive overview of the literature pertaining to relationship management” and that ii) “this seemingly perfectly matches with the results, which is odd - especially in light of the qualitative research goals and the exploratory nature of the research”. In response, the authors state that “the present study is not about conflict resolution but about managing relationship problems” and “it would not be appropriate to expand on conflict resolution as a theoretical framework”. This, however, does not answer my comments. Why are strategies “proposed” – which match seemingly perfectly with the results? My point is not that the study is seemingly about conflict resolution, but that, the novelty of the work (in light of existing studies) could be made much clearer upfront. The addition to section 1.4 is beneficial in this respect, but would be more convincing if it was stated at the beginning of section 1.3.
Yes, we now see your point. Following your suggestion, we have added a paragraph at the beginning of the section 1.3 (second paragraph) where we make the distinction between conflict resolution and strategies for managing relationship problems.
A similar issue pertains to the discussion section. On the one hand, the authors state that no hypotheses were formulated due to this being exploratory research. However, in their response letter, the authors state “Our argument, as stated in the introduction, is that we can draw a number of hypotheses about the strategies people are likely to use in order to manage relationship problems, yet because the phenomenon is complex and there is limited research in the area, we cannot predict all these strategies, so our study was exploratory”. Why remain so intent on the use of hypotheses? Deductive research builds on existing theory and from that, deduces hypotheses. Inductive research, does the opposite: explorative work to produce theories. Considering that the authors also answer “To best of our knowledge, there is no study that has attempted to identify these strategies, which we consider an important omission in the literature that the current study aims to address”, it simply feels forced to want to portray this work as both exploratory and responsive to hypotheses. I understand your aim, but I suggest adhering completely to the exploratory strength of the paper.
We see your point; however, our work is both exploratory and hypothesis-driven. If it were solely hypothesis-driven, we would focus on testing the presence of strategies we have derived. Conversely, if it were purely exploratory, we would not use any theoretical framework to derive hypotheses. Here, we have combined these approaches. Specifically, we employed an evolutionary theoretical framework to guide us in identifying possible strategies that people might use to manage relationship problems; however, our analysis was not confined to these strategies. Our rationale for this approach is that the theoretical framework we adopted was unlikely to predict all possible strategies people might use to manage relationship problems. On the other hand, using only an exploratory approach without a guiding theoretical framework would limit our understanding of the identified strategies. Generally, we believe that a phenomenon is better understood within the context of a solid theoretical framework that does not limit inquiry. We understand your concern, as most studies typically adopt one approach or the other, but we do not believe that all studies need to adhere to only one approach.
It was found that older participants were less likely to use “communication and problem solving” as a strategy than younger participants. The authors answer that “it is not very clear why older people would be less adept at communication than younger ones. It could be argued that older people are more experienced and thus, are better in communicating with their partners”. I still expect reflection on this in the discussion. Emotional avoidance, the perception of issues and established (non-)communicative patterns in this respect are all valid and well-discussed explanations that can be found in the literature. Some examples:
Zietlow, P. H., & Sillars, A. L. (1988). Life-stage differences in communication during marital conflicts. Journal of Social and Personal Relationships, 5(2), 223-245.
Henry, R. G., Miller, R. B., & Giarrusso, R. (2005). Difficulties, disagreements, and disappointments in late-life marriages. The International Journal of Aging and Human Development, 61(3), 243-264.
Jensen, J. F., & Rauer, A. J. (2015). Marriage work in older couples: Disclosure of marital problems to spouses and friends over time. Journal of Family Psychology, 29(5), 732.
Thank you very much for the references. Following your suggestion, we now discuss them in the text (lines 469-477).
Figure 1 was added to the manuscript, but the caption refers to “willingness” to use strategies – which, throughout the rest of the manuscript, the authors have changed to “likelihood”.
Yes good point. We have changed “willingness” to “likelihood.”
